# Hypotony after 25-gauge pars plana vitrectomy with suturing all sclerotomies

**Shogo Takahashi**[☉], **Tomoko Ueda-Consolvo**[iD][☉]*, **Yuta Takata, Masaaki Ishida, Shuichiro Yanagisawa, Atsushi Hayashi**

Department of Ophthalmology, Graduate School of Medicine and Pharmaceutical Sciences, University of Toyama, Toyama, Japan

☉ These authors contributed equally to this work.
* ueda@med.u-toyama.ac.jp

## Abstract

### Purpose

To evaluate the incidence of postoperative hypotony and early complications in 25-gauge vitrectomy by suturing all sclerotomies.

### Methods

We retrospectively reviewed a consecutive series of 458 eyes of 435 patients who underwent 25G pars plana vitrectomy with suturing all sclerotomies. The primary outcome measure was intraocular pressure (IOP) at postoperative day 1. Secondary outcome measures were clinical signs of hypotony and postoperative complications. Hypotony was defined as an IOP < 5 mmHg.

### Results

Postoperative hypotony was found in 22 eyes (4.8%). No eyes suffered IOP of 0 nor 1 mmHg. None of the eyes demonstrated any clinical signs of hypotony including endophthalmitis, hypotony maculopathy nor choroidal detachment. The eyes with sulfur hexafluoride (SF6) gas tamponade had a significantly lower rate of hypotony compared to eyes with no tamponade (1.3% (2 eyes/154 eyes) vs 6.3% (18eyes/284 eyes); p = 0.0118). The incidence of hypotony was 11.1% (5 eyes/45 eyes) in reoperation and 4.1% (17 eyes/413 eyes) in primary operations (p = 0.037).

### Conclusion

Suturing all sclerotomies showed a low incidence of postoperative hypotony. Reoperation increased the risk of hypotony, while SF6 gas tamponade lowered the risk of hypotony.

**Data availability statement:** All relevant data are within the manuscript and its Supporting Information files.

**Funding:** The author(s) received no specific funding for this work.

**Competing interests:** The authors have declared that no competing interests exist.

## Introduction

In the medical world, safe, minimally invasive treatment is always desired, and vitreous surgery has made remarkable progress. A 25-gauge (G) system for small incision sutureless vitrectomy was introduced in 2002 [1,2] followed by a 27G system in 2010 [3]. These techniques cause less postoperative inflammation and faster visual recovery compared to 20G vitrectomy [4]. Currently, microincision vitrectomy surgery (MIVS) is mainstream. MIVS sclerotomies are expected to close without suturing; it has been reported that the smaller the scleral wound, the shorter the time required for wound closure [5–9]. Even with small incisions of 25G or 27G, however, some patients suffered postoperative hypotony [9–13] because of inadequate closure of sclerotomies which caused slow leakage. Strategies to lower the risk of severe hypotony are indispensable because it might lead to irreversible visual impairment due to maculopathy, endophthalmitis and choroidal detachment. Suturing every sclerotomy may reduce the risk of hypotony, but the real-world incidence of hypotony in 25 G vitrectomy with suturing remains unexamined. The purpose of this study is to investigate the incidence of postoperative hypotony and early complications in 25 G vitrectomy in which all sclerotomies were sutured at the end of the operation.

## Methods

We retrospectively reviewed a consecutive series of 458 eyes of 435 patients who underwent 25 G pars plana vitrectomy performed by 5 surgeons (TUC, MI, SA, SY, AH) at Toyama University Hospital between October 2022 and September 2023. The study was approved by the Institutional Review Board of the University of Toyama (approval no. R2024055), and the procedures used conformed to the tenets of the Declaration of Helsinki. Informed consent was obtained in the form of opt-out on the website. Access for data collection was conducted between June 25, 2024 and August 31, 2024. All patient records and data were anonymized and de-identified. Surgical indications were epiretinal membrane (ERM) (144 eyes), rhegmatogenous retinal detachment (RRD) (112 eyes), macular hole (MH) (46 eyes), vitreous hemorrhage (VH) (45 eyes), intraocular lens (IOL) dislocation (34 eyes), proliferative diabetic retinopathy (PDR) (15 eyes), retinoschisis (14 eyes), vitreous opacities (9 eyes), vitreomacular traction syndrome (VMTs) (6 eyes), subretinal hemorrhage (SRH) (5 eyes), lens subluxation (7 eyes), and asteroid hyalosis (1 eye).

Patient medical records were reviewed, and the following data were collected: age, sex, the side of operation (OD or OS), preoperative and postoperative intraocular pressure (IOP), operating time, tamponade materials (balanced salt solution (BSS), air, sulfur hexafluoride ($SF_6$) or perfluoro propane ($C_3F_8$)), results of anterior segment and fundus examinations including optical coherence tomography (OCT) (RS-3000 Advance; NIDEK Co, Ltd, Aichi, Japan or DRI-OCT Toriton; Topcon Inc, Tokyo, Japan), and complications within one month after surgery. Preoperative IOP was measured using a noncontact tonometer (FT-01 Tomey, Nagoya, Japan), and postoperative IOP was measured with a handheld rebound tonometer (iCare Pro; Icare Finland Oy, Helsinki, Finland). Hypotony was defined as an IOP < 5 mmHg. Cases of glaucoma implant insertion and silicone oil injection were excluded.

## Vitrectomy

All eyes underwent a standard pars plana vitrectomy with three 25-gauge ports. Anesthesia, either local or general, depended on the patient's preference. Vitrectomy was performed using either the EVA (DORC, Zuidland, The Netherlands) or the Constellation (Alcon Laboratories, Inc., Fort Worth, TX). The trocars for transscleral cannulas were placed through the pars plana in the superonasal, superotemporal and inferotemporal quadrants for the site of infusion and insertion of intraocular instruments. Each sclerotomy was made with an oblique 30° incision through the sclera. All scleral wounds, regardless of their leakage, were sutured using 10−0 or 9−0 absorbable sutures, and no pressure was applied to the wounds. The planned suture removal was not performed after surgery.

## Statistical analysis

All statistical analyses were carried out using JMP statistical discovery software (version 17; SAS Institute, Cary, NC, USA). Mixed-effects models were applied to compare both continuous variables (age, operating time, preoperative IOP) and categorical variables (sex, side of operation (OD or OS), previous vitrectomy, gas usage for tamponade) between the eyes with hypotony and those without. Pearson's correlation procedure was used to investigate correlations between data including age, preoperative IOP, operating time, and postoperative IOP. Multivariable logistic regression analysis was performed for postoperative hypotony, including as covariates: reoperation status, types of tamponade (none/ air/ SF6/ C3F8), combined phacoemulsification, operating time, preoperative IOP, age and sex. Statistical significance was defined as $P < 0.05$.

# Results

## Patients

Four hundred fifty-eight eyes of 435 patients were included. The mean age was $65.8 \pm 11$ years, and 261 patients (57%) were men. In 283 eyes (62%), vitrectomy was combined with phacoemulsification and aspiration with IOL implantation. The indication for 25G vitrectomy is presented in Table 1. In the 174 tamponade cases (38%), 20% $SF_6$ was used in 154 cases (33.6%), 12% $C_3F_8$ gas was used in 5 cases (1.1%), and air was used in 15 cases (3.3%) (Table 1). Overall, mean preoperative intraocular pressure was $15.8 \pm 6.2$ mmHg. The mean operating time was $63.1 \pm 33$ minutes, and the mean postoperative IOP was $14.6 \pm 8.4$ mmHg.

## Hypotony

Hypotony was found in 22 eyes (4.8%) on postoperative day 1; 2 mmHg in 4 eyes (18%), 3 mmHg in 7 eyes (32%), 4 mmHg in 11 eyes (32%), and none of the eyes suffered IOP of 0 nor 1 mmHg (Table 2). The mean IOP in hypotony eyes was $3.3 \pm 0.76$ mmHg (median, 3.5 mmHg). The incidence of hypotony was 8.3% in ERM surgery (12 eyes/144 eyes), 0.8% in RRD surgery (1 eye/112 eyes), 2.2% in MH surgery (1 eye/46 eyes), 4.4% in VH surgery (2 eyes/45 eyes), 2.9% in IOL dislocation surgery (1 eye/34 eyes), and 7.1% in retinoschisis surgery (1eye/14 eyes) (Table 1). None of the 22 eyes demonstrated hypotony maculopathy nor choroidal detachment on OCT. At the follow-up within one week after surgery, the intraocular pressure had improved to above 5 mmHg in all the 22 eyes, and there were no abnormal findings associated with hypotony on OCT. Eyes with 20%SF6 tamponade had a significantly lower incidence of hypotony compared to those with no tamponade (p=0.0118; 95%CI: −0.094 to −0.012). A remarkable difference in the risk for hypotony was found between primary operations and reoperations (p=0.0367; 95%CI: 0.0043 to 0.13). No significant differences were found in sex, age, the side of operation (OD or OS), mean operating time, or preoperative IOP between hypotony group and non-hypotony group (p=0.573, p=0.263, p=0.182, p=0.479, p=0.996, respectively) (Table 1). There was a weak correlation between operating time and postoperative IOP (r=0.214; 95%CI: 0.12 to 0.30; p<0.0001). There was no correlation between age and postoperative IOP (r=−0.0632; 95%CI: −0.15 to 0.029; p=0.177). The correlation between

**Table 1. Hypotony After 25-gauge Vitrectomy with suturing all sclerotomies.**

| | Hypotony, n(%) |
|---|---|
| Total eyes (n = 458) | 22 (4.8) |
| **indication** | |
| ERM (n = 144) | 12 (8.3) |
| RRD (n = 112) | 1 (0.8) |
| MH (n = 46) | 1 (2.2) |
| VH (n = 45) | 2 (4.4) |
| IOL dislocation (n =34) | 1 (2.9) |
| PDR (n = 15) | 0 (0) |
| retinoschisis (n = 14) | 1 (7.1) |
| vitreous opacities (n = 9) | 0 (0) |
| VMTs (n = 6) | 1 (16.6) |
| SRH (n = 5) | 1 (20.0) |
| asteroid hyalosis (n =1) | 1 (100) |
| lens subluxation (n = 7) | 1 (14.3) |
| others (n = 20) | 0 (0) |
| **Eye** | P = 0.535 [a] (95%CI: -0.052, 0.027) |
| OD (n = 235) | 10 (4.3) |
| OS (n = 223) | 12 (5.4) |
| **Sex** | P = 0.294 [a] (95%CI: -0.018, 0.061) |
| male (n = 261) | 10 (3.8) |
| female (n =197) | 12 (6.0) |
| **No/ 20%SF6 tamponade** | **P = 0.0118*[a]** (95%CI: -0.094, -0.012) |
| No tamponade (n = 284) | 18 (6.3) |
| 20%SF6 (n = 154) | 2 (1.3) |
| Air (n = 15) | 2 (13.3) |
| 12%C3F8 (n = 5) | 0 (0) |
| **Reoperation** | **P = 0.0367*[a]** (95%CI: 0.0043, 0.13) |
| Reoperation (n = 45) | 5 (11.1) |
| Primary operation (n = 413) | 17 (4.1) |
| **Mean age (years)** | P = 0.146 [a] (95%CI: -1.27, 8.60) |
| 65±11.5 | 68.0±9.94 |
| **preoperative IOP (mmHg)** | P = 0.948 [a] (95%CI: -2.62, 2.80) |
| 15.8±6.2 | 15.9±5.9 |
| **Mean operating time (minutes)** | P = 0.472 [a] (95%CI: -19.5, 9.05) |
| 63.0±32.2 | 58.3±42.4 |

Data are presented as means±SD or n (%).

ERM, epiretinal membrane; RRD, rhegmatogenous retinal detachment; MH, macular hole; VH, vitreous hemorrhage; IOL, intraocular lens; PDR, proliferative diabetic retinopathy; VMTs, vitreomacular traction syndrome; SRH, subretinal hemorrhage; SF6, sulfur hexafluoride; C3F8, perfluoropropane; IOP, intra-ocular pressure; 95%CI, 95% confidence interval, a: mixed-effects models, **\*Statistical significance was defined as P<0.05.

Data are presented as means±SD or n (%). ERM: epiretinal membrane, RRD: rhegmatogenous retinal detachment, MH: macular hole, VH: vitreous hemorrhage, PDR: proliferative diabetic retinopathy, VMTs: vitreomacular traction syndrome, SRH: subretinal hemorrhage

a: Student's t-test, b: Chi-square test

**Table 2. Characteristics of cases of hypotony.**

| Case No. | sex | Age (years) | indication | surgical procedure | operating time (min) | reopera-tion | preoperative IOP (mmHg) | Air/gas tamponade | postoperative IOP (mmHg) | compli-cations |
|---|---|---|---|---|---|---|---|---|---|---|
| 1 | M | 73 | SRH | PPV | 35 | no | 10 | air | 2 | no |
| 2 | M | 45 | VH | tri | 80 | no | 15 | air | 2 | VH |
| 3 | F | 75 | ERM | tri | 40 | no | 14 | no | 2 | no |
| 4 | F | 91 | lens subluxation | PPV+ intrascleral fixation | 199 | no | 12 | no | 2 | no |
| 5 | M | 72 | ERM | PPV | 120 | no | 18 | no | 3 | no |
| 6 | M | 69 | ERM | PPV | 19 | yes | 11 | no | 3 | no |
| 7 | M | 78 | MH | tri | 96 | no | 11 | 20%SF6 | 3 | no |
| 8 | F | 70 | ERM | PPV | 17 | no | 14 | no | 3 | no |
| 9 | F | 66 | ERM | PPV | 21 | no | 15 | no | 3 | no |
| 10 | F | 79 | ERM | PPV | 20 | no | 17 | no | 3 | no |
| 11 | F | 64 | IOL dislocation | IOL extraction+PPV+IOL intrascleral fixation | 84 | no | 17 | no | 3 | no |
| 12 | M | 71 | ERM | tri | 63 | no | 16 | no | 4 | no |
| 13 | M | 63 | ERM | tri+trabeculotomy | 35 | yes | 13 | no | 4 | no |
| 14 | M | 75 | VH | PPV+IOL | 54 | yes | 30 | no | 4 | no |
| 15 | M | 62 | RRD | PPV | 91 | no | 7 | 20%SF6 | 4 | no |
| 16 | M | 77 | asteroid hyalosis | tri | 21 | no | 12 | no | 4 | no |
| 17 | F | 57 | ERM | tri | 63 | no | 33 | no | 4 | no |
| 18 | F | 62 | ERM | tri | 36 | no | 16 | no | 4 | no |
| 19 | F | 82 | ERM | tri | 46 | no | 13 | no | 4 | no |
| 20 | F | 67 | ERM | tri | 67 | no | 17 | no | 4 | no |
| 21 | F | 61 | VMTs | PPV | 36 | yes | 16 | no | 4 | no |
| 22 | F | 58 | retinoschisis | PPV | 40 | yes | 15 | no | 4 | no |

M, male; F,female; SRH, subretinal hemorrhage; PPV, pars plana vitrectomy; tri, PPV combined with cataract surgery (phacoemulsification and aspiration with intraocular lens implantation); VH, vitreous hemorrhage; ERM, epiretinal membrane; MH, macular hole; IOL, intraocular lens implantation; RRD, rhegmatogenous retinal detachment; VMTs, vitreomacular traction syndrome; IOP, intraocular pressure; SF6, sulfur hexafluoride.

preoperative IOP and postoperative IOP was statistically significant (r = 0.189; 95%CI: 0.099 to 0.28; p < 0.0001), but the effect size was small and not clinically meaningful. In the multivariable logistic regression analysis, reoperation was significantly associated with postoperative hypotony (odds ratio = 4.28; 95% CI:1.20 to 15.3; p = 0.025) and 20%SF6 tamponade was associated with a significantly lower risk compared to no tamponade (odds ratio = 0.16; 95% CI: 0.033 to 0.79; p = 0.025) (Table 3).

## Postoperative complications

Postoperative complications except for hypotony were observed in 27 eyes (5.9%); vitreous hemorrhage in 12 eyes (2.4%), macular edema in 4 eyes (0.87%), RRD in 3 eyes (0.66%), IOL dislocation in 2 eyes (0.44%), ERM in 1 eye (0.21%), and re-detachment of RRD was observed in 5 eyes (4.4%) (Table 4). Re-detachment of RRD occurred in one eye on postoperative day 10, in one eye at 2 weeks, in one eye at 3 weeks and in 2 eyes at 4 weeks postoperatively. No eyes suffered hypotony maculopathy, endophthalmitis, nor choroidal detachment.

## Discussion

In the present study, the incidence of postoperative hypotony was 4.8% and no complications associated with hypotony including endophthalmitis, hypotony maculopathy nor choroidal detachment occurred. Our study clearly showed that

**Table 3. Multivariable logistic regression analysis of risk factors for postoperative hypotony.**

| Variable | Odds ratio | 95% CI | P-value |
|---|---|---|---|
| Reoperation | 4.28 | 1.20-15.3 | 0.025* |
| Type of tamponade | | | |
| None (Reference) | 1.00 | | |
| 20%SF6 | 0.16 | 0.033-0.79 | 0.025* |
| Air | 2.66 | 0.53-13.4 | 0.23 |
| 12%C3F8 | Not estimable† | Not estimable† | – |
| Combined phacoemulsification | 0.71 | 0.26-1.91 | 0.49 |
| Operating time | 0.99 | 0.98-1.01 | 0.44 |
| Preoperative IOP | 1.03 | 0.97-1.12 | 0.47 |
| Age | 0.98 | 0.94-1.02 | 0.46 |
| Sex, male(reference) | 1.00 | | |
| female | 1.77 | 0.71-4.43 | 0.22 |

95% CI, 95% confidence interval; SF6, sulfur hexafluoride; C3F8, perfluoropropane; IOP, intraocular pressure, *Statistical significance was defined as P<0.05, †OR and 95% CI were not estimable due to extremely low event frequency.

**Table 4. Postoperative complications.**

| VH | 12 (2.4) |
|---|---|
| ME | 4 (0.87) |
| RRD | 3 (0.66) |
| IOL dislocation | 2 (0.44) |
| ERM | 1 (0.21) |
| Re-detachment in RRD eyes | 5 (4.4) |

Data are presented as n (%).

VH, vitreous hemorrhage; ME, macular edema; RRD, rhegmatogenous retinal detachment; IOL, intraocular lens; ERM, epiretinal membrane

suturing sclerotomy had a low incidence of irreversible visual impairment due to severe hypotony in MIVS. We also found that reoperation increased the risk of hypotony, while air or gas tamponade lowered the risk of hypotony.

Small-gauge sutureless vitrectomy has reduced operating time, surgically induced astigmatism and discomfort from conjunctival sutures, and achieved faster visual improvement. Postoperative hypotony after sutureless vitrectomy, however, remains a cause for concern. O' Reilly and Beatty S reported that 10 eyes out of 39 eyes (25.6%) suffered transient hypotony (IOP<5 mmHg) after 25-gauge transconjunctival sutureless vitrectomy [14]. To reduce the risk of leakage, they recommended lateral displacement of the conjunctiva when inserting the microcannulae to cover the sclerotomies. Uy HS and colleagues found that 11 eyes out of 55 eyes (22%) had postoperative hypotony (IOP<5 mmHg) using 25-gauge, beveled-tip, sutureless vitrectomy [15]. Postoperative day 1 IOP was 0 mmHg in 2 eyes (3.6%), and 3 eyes (6%) had adverse events due to hypotony. These studies indicate that not a few sclerotomies need to be sutured for proper wound closure even in small-gauge vitrectomy. Bamonte G and colleagues performed sutureless vitrectomy except in cases in which they found fluid or gas leakage; they sutured the leaking sclerotomy with a single stitch of 9−0 vicryl [16]. In 8 out of 122 eyes, sclerotomy was sutured and none of them developed hypotonus after surgery. Their incidence of postoperative hypotony (IOP≤5 mmHg) was 13.1% (16 eyes/122 eyes) and day 1 IOP was 0 mmHg in 2 eyes. Our study found that the incidence of postoperative hypotony was lower; 34 eyes (7.4%) had a postoperative IOP of 5 mmHg or less, and no eyes

suffered IOP of 0 nor 1 mmHg. These results suggest that physicians cannot find every slow leakage covered with conjunctiva, and that suturing all sclerotomies is more secure than simply suturing as needed.

Air or gas tamponade is also known to lower the risk of hypotony [5,16]. Bamonte G and colleagues reported that eyes with air or gas tamponade had a significantly lower rate of hypotony compared to eyes with no tamponade (3.3% vs 22.4%; p = 0.002) [16]. Bourgault S and colleagues also showed that air or gas tamponade resulted in a 4-fold reduction in the incidence of hypotony after 25-gauge transconjunctival sutureless vitrectomy [17]. The incidence of hypotony was 4.8% in eyes with air or gas tamponade and 20.0% in eyes filled with fluid (p = 0.0001). Consistent with their reports, we found a significant difference in the rate of hypotony; 2.3% in eyes with air or gas tamponade and 6.3% in eyes with no tamponade (p = 0.0498). The frequency of hypotony was lower than their studies. In other words, suturing sclerotomy even lowers the incidence of hypotony in eyes with air or gas tamponade.

Reoperations, on the other hand, predisposed eyes to hypotony. In the present study, the incidence of hypotony was 11.1% (5 eyes/45 eyes) in reoperation and 4.1% (17 eyes/413 eyes) in primary operations (p = 0.037). These results were comparable to previous studies on sutureless vitrectomy. Shimada H and colleagues reported that leakage occurred in 2 of 298 eyes (1%) in primary operations and 3 of 39 eyes (8%) in reoperation (p = 0.0006) [18]. Bamonte G also showed that 9 eyes out of 98 eyes (9.2%) had hypotony in primary operation, while 7 eyes out of 24 eyes (29.2%) had hypotony in reoperation (p = 0.009) [16]. It is possible that eyes in reoperations may have less vitreous left, resulting in less internal vitreous plugging in sclerotomies.

In our study, the rate of postoperative complications was comparable or better than that in previous studies; vitreous hemorrhage was 2.4% in our study and 5–8% [9,11,19] in previous studies; macular edema, 0.87% vs 7–10% [19–21]; and rhegmatogenous retinal detachment, 0.66% vs 0.43–2% [22–38]. Considering the low incidence of hypotony in our study, 25-gauge vitrectomy with suturing sclerotomies may lower the overall risk of complications and may even be superior to sutureless vitrectomy because hypotony could cause irreversible visual impairment. Suturing sclerotomies is, however, not without issues. Discomfort from sutures, surgically induced astigmatism and longer operating time are troublesome. Suturing every sclerotomy may be overtreatment in the majority of cases. Severe hypotony is, however, one of the critical adverse events for visual prognosis. Physicians should carefully consider the risk of postoperative hypotony in each eye and determine whether suturing sclerotomies would improve the outcome or not. In addition, Discomfort and refractive changes should also be considered when evaluating the necessity of suturing.

The retrospective nature, a non-comparative study, and inclusion of various vitreoretinal diseases potentially limit the conclusions of this study. Some limitations were intrinsic to the 5 surgeons because of different preferences and techniques. Further investigations are warranted to confirm our observations. Despite these limitations, we believe our findings indicate that suturing sclerotomies is a better treatment option for eyes in reoperations and especially when not using gas tamponade.

In conclusion, suturing every sclerotomy had no incidence of irreversible visual impairment due to severe hypotony in MIVS. The risk of hypotony was increased by reoperation and lowered by gas tamponade.

## Supporting information

**S1 File. All data included in the analysis.**
(XLSX)

## Author contributions

**Data curation:** Shogo Takahashi, Tomoko Ueda Consolvo, Yuta Takata, Masaaki Ishida, Shuichiro Yanagisawa, Atsushi Hayashi.

**Formal analysis:** Shogo Takahashi.

**Methodology:** Shogo Takahashi, Tomoko Ueda Consolvo, Atsushi Hayashi.

**Supervision:** Tomoko Ueda Consolvo, Atsushi Hayashi.

**Writing – original draft:** Shogo Takahashi.

**Writing – review & editing:** Tomoko Ueda Consolvo.

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
