## [Decision Letter · Decision Letter 0]

13 Oct 2025

Dear Dr. Consolvo,

Thank you for submitting your manuscript to PLOS ONE. After careful consideration, we feel that it has merit but does not fully meet PLOS ONE’s publication criteria as it currently stands. Therefore, we invite you to submit a revised version of the manuscript that addresses the points raised during the review process.

Significant issues have been raised regarding the absence of a study control group in which sutures were not used, therefore preventing acceptance of the concluding claim that sutured sclerotomies reduce hypotony. Also, there are concerns about the study design, which assessed hypotony only on the first postoperative day and lacked longitudinal follow-up. This eliminates some patients with a clinical subset of hypotony. Moreover, the technique of intraocular pressure evaluation was not thoroughly described. This is essential for the study's reproducibility. May I invite you to respond to these comments, including your concerns about the generalizability of the study's findings?

We look forward to receiving your revised manuscript.

Kind regards,

Ogugua Ndubuisi Okonkwo, M.D.

Academic Editor

PLOS ONE

2. In the online submission form, you indicated that [The datasets are available from Tomoko Ueda-Consolvo, the correponding author, on reasonable request.].

Additional Editor Comments (if provided):

Reviewers' comments:

Reviewer's Responses to Questions

**Comments to the Author**

1. Is the manuscript technically sound, and do the data support the conclusions?

Reviewer #1: Partly

Reviewer #2: No

2. Has the statistical analysis been performed appropriately and rigorously?

Reviewer #1: Yes

Reviewer #2: Yes

3. Have the authors made all data underlying the findings in their manuscript fully available?

Reviewer #1: Yes

Reviewer #2: Yes

4. Is the manuscript presented in an intelligible fashion and written in standard English?

Reviewer #1: Yes

Reviewer #2: Yes

Reviewer #1: Dear Authors,

I had the pleasure your work evaluating the incidence of postoperative hypotony in 25-gauge vitrectomy by suturing all sclerotomies. The abstract is well structured. The introduction is focused on the topic. The strenght of this scientific work is the large consecutive cohort.

However, the manuscript needs a deep review:

-I would suggest to change "Our 130 study clearly showed that suturing sclerotomy reduces the incidence of irreversible visual impairment due 131 to severe hypotony in MIVS." (line 129) because the design is retrospective and not comparative, so this statement appears unbalanced.

-I would also point out that there is a discrepancy in numbers: eyes with ERM (line 58: 144 eyes; line 101: 114 eyes), eyes with RRD (line 59, line 101)

-Then I noticed that the conclusion "Eyes with air or gas tamponade had a significantly 104 lower incidence of hypotony compared to those with no tamponade (p=0.0498)." appears to be an artifact. In fact, air has 13.3% of hypotony, while no tamponade has 6.3%. Please change accordingly.

-Please specify how long it was the follow up in "Re-detachment in RRD eyes 5 (4,4%)"

-Insufficient statistical analysis: PLOS requires reporting of effect estimates with 95% confidence intervals, not p-values alone.

-Please perform a multivariable logistic regression for postoperative hypotony, including as covariates: reoperation status, type of tamponade (none/air/SF6/C3F8), surgical indication, combined phacoemulsification (yes/no), operative time, preoperative IOP, age, and sex.

-I noticed that there are 458 eyes and 438 patients: please verify if a inter-eye bias does exist. I suggest a mixed model to address this potential bias.

-Misspelling: "guage"  gauge "retionoschisis" "retinoschisis"

Reviewer #2: I have reviewed your manuscript on “Incidence of Postoperative Hypotony and Early Complications in 25-Gauge Vitrectomy by Suturing All Sclerotomies.” The study addresses an important and clinically relevant topic; however, I would like to raise a few methodological concerns that may limit the strength of your conclusions. The manuscript may be more suitable for consideration as a Brief Report or Short Communication, rather than a full original article.

1. The absence of a comparative control group (e.g., cases without suturing of sclerotomies) makes it difficult to determine the true impact of universal suturing on the rate of postoperative hypotony. Without direct comparison, the claim that suturing reduces hypotony remains speculative.

2. The study evaluates intraocular pressure only on postoperative day 1. Since hypotony may develop or resolve over time, the lack of longitudinal follow-up data limits the ability to understand the persistence or clinical relevance of hypotony beyond the immediate postoperative period.

3. The manuscript does not clearly specify the instrument and method used for intraocular pressure assessment. Given that small variations in measurement technique (e.g., Goldmann applanation vs. non-contact tonometry vs. rebound tonometry) can significantly influence IOP values, clarification is essential for the validity of the results.

4. The paper does not elaborate on whether conjunctival displacement or beveled versus straight insertion techniques were used. These factors are known to influence wound closure and postoperative leakage risk. Their omission makes it difficult to generalize the findings or compare them with prior literature.

5. Being a retrospective review from one institution, the results may reflect local surgical techniques, case mix, or surgeon preference rather than generalizable outcomes.

6. The series included a wide variety of vitreoretinal pathologies (ERM, RRD, MH, VH, PDR, etc.), yet the outcomes were pooled together. Since different conditions may carry different risks of postoperative hypotony, this heterogeneity makes interpretation less clear.

7. Hypotony was defined strictly as IOP < 5 mmHg on day 1. While this is a common definition, functional hypotony (associated with maculopathy or choroidal changes) or transient hypotony beyond day 1 is not captured.

8. The study focuses almost exclusively on IOP, but does not address visual acuity, refractive change, or patient-reported outcomes (such as suture-related discomfort), which are relevant when recommending routine suturing. The study focuses almost exclusively on IOP, but does not address visual acuity, refractive change, or patient-reported outcomes (such as suture-related discomfort), which are relevant when recommending routine suturing.

9. Given the lack of a control group, limited time point, and retrospective design, the conclusion that suturing "reduces irreversible visual impairment" is overstated and not directly supported by the presented evidence.

**Do you want your identity to be public for this peer review?** For information about this choice, including consent withdrawal, please see our Privacy Policy

Reviewer #1: No

Reviewer #2: **Yes:** Hamid Riazi-Esfahani

---

## [Author Response · Author response to Decision Letter 1]

23 Nov 2025

Response to Reviewers has been uploaded as a separate file.

---

## [Decision Letter · Decision Letter 1]

15 Jan 2026

Hypotony after 25-gauge pars plana vitrectomy with suturing all sclerotomies

PONE-D-25-37877R1

Dear Dr. Consolvo,

We’re pleased to inform you that your manuscript has been judged scientifically suitable for publication and will be formally accepted for publication once it meets all outstanding technical requirements.

Kind regards,

Ogugua Ndubuisi Okonkwo, M.D.

Academic Editor

PLOS One

Reviewers' comments:

Reviewer's Responses to Questions

**Comments to the Author**

Reviewer #1: All comments have been addressed

Reviewer #2: All comments have been addressed

2. Is the manuscript technically sound, and do the data support the conclusions?

Reviewer #1: Yes

Reviewer #2: Yes

3. Has the statistical analysis been performed appropriately and rigorously?

Reviewer #1: Yes

Reviewer #2: Yes

4. Have the authors made all data underlying the findings in their manuscript fully available?

Reviewer #1: Yes

Reviewer #2: Yes

5. Is the manuscript presented in an intelligible fashion and written in standard English?

Reviewer #1: Yes

Reviewer #2: Yes

Reviewer #1: Dear Authors,

I appreciated that you adapted the manuscript according my suggestions. With these adjustements, in my opinion, the paper Is suitable for publication. Thank you for you work.

Reviewer #2: I believe that the revisions have strengthened the scientific rigor, improved the clarity of presentation, and aligned the discussion and conclusions more closely with the available evidence. In its revised form, I feel that the manuscript now adequately addresses the reviewers’ concerns and is suitable for publication.

**Do you want your identity to be public for this peer review?** For information about this choice, including consent withdrawal, please see our Privacy Policy

Reviewer #1: No

Reviewer #2: **Yes:** Hamid Riazi-Esfahani

---

## [Editor Report · Acceptance letter]

PONE-D-25-37877R1

PLOS One

Dear Dr. Consolvo,

I'm pleased to inform you that your manuscript has been deemed suitable for publication in PLOS One. Congratulations! Your manuscript is now being handed over to our production team.

Kind regards,

on behalf of

Prof Ogugua Ndubuisi Okonkwo

Academic Editor

PLOS One